# SDFPoseGraphNet: Spatial Deep Feature Pose Graph Network for 2D Hand Pose Estimation

**DOI:** 10.3390/s23229088

**Published:** 2023-11-10

**Authors:** Sartaj Ahmed Salman, Ali Zakir, Hiroki Takahashi

**Affiliations:** 1Department of Informatics, Graduate School of Informatics and Engineering, The University of Electro-Communications, Tokyo 182-8585, Japan; a2240012@edu.cc.uec.ac.jp (A.Z.); rocky@inf.uec.ac.jp (H.T.); 2Artificial Intelligence Exploration Research Center/Meta-Networking Research Center, The University of Electro-Communications, Tokyo 182-8585, Japan

**Keywords:** 2D HPE, VGG-19, Spatial Attention, HCI, VR, CPM

## Abstract

In the field of computer vision, hand pose estimation (HPE) has attracted significant attention from researchers, especially in the fields of human–computer interaction (HCI) and virtual reality (VR). Despite advancements in 2D HPE, challenges persist due to hand dynamics and occlusions. Accurate extraction of hand features, such as edges, textures, and unique patterns, is crucial for enhancing HPE. To address these challenges, we propose SDFPoseGraphNet, a novel framework that combines the strengths of the VGG-19 architecture with spatial attention (SA), enabling a more refined extraction of deep feature maps from hand images. By incorporating the Pose Graph Model (PGM), the network adaptively processes these feature maps to provide tailored pose estimations. First Inference Module (FIM) potentials, alongside adaptively learned parameters, contribute to the PGM’s final pose estimation. The SDFPoseGraphNet, with its end-to-end trainable design, optimizes across all components, ensuring enhanced precision in hand pose estimation. Our proposed model outperforms existing state-of-the-art methods, achieving an average precision of 7.49% against the Convolution Pose Machine (CPM) and 3.84% in comparison to the Adaptive Graphical Model Network (AGMN).

## 1. Introduction

The rapid advancement of deep neural networks has meant significant progress in hand pose estimation (HPE), a critical component in AI-driven applications such as human–computer interaction (HCI) and virtual reality (VR). However, HPE remains a complex task due to the flexibility of hand joints, local similarities, and severe occlusions, including self- and object occlusion [1]. Therefore, it is crucial to investigate efficient HPE architectures that effectively address these challenges [2]. Over the past decade, numerous 2D methods, including multiview RGB models [3,4], depth-based architectures [5,6,7], and monocular RGB methods [8,9], have been developed in the HPE field to overcome these problems. More recently, 3D HPE has acquired increased interest due to its enhanced accuracy and performance.

Nevertheless, 2D HPE remains a vital research direction, as it forms the foundational building block for 3D algorithms, which achieve estimation results by transposing feature maps from 2D to 3D space [10,11]. Reflecting the importance of 2D HPE from a monocular RGB image is a central point of the experiment. In HPE, extracting valuable feature maps is a vital and challenging task for precisely identifying and tracking the position and orientation of keypoints of each joint of the hand in each scene. These feature maps include edges, corners, textures, or other complex patterns unique to hand structures. These features serve as the input to deep learning (DL) models, allowing them to learn the spatial and temporal relationships among various joints and accurately localize the coordinates of each joint. Enhanced versions of features impact the accuracy of HPE models, even in challenging situations such as self- or object occlusion, varying lighting conditions, or frequent hand movements.

With the advent of DL, the Convolutional Pose Machine (CPM) [12] emerged as a pioneering model for human pose estimation, and it has also become a significant model in the realm of 2D HPE. CPM can learn robust feature maps but frequently struggles to comprehend the geometric constraints among joints. This often leads to inconsistencies in the final joint predictions, a particularly noticeable challenge in human pose estimation tasks. The issue can be even more pronounced in 2D HPE due to the increased articulation and severity of self-occlusion. To better understand the interconnections among joints, various research work has investigated the potential of integrating deep convolutional neural networks (DCNNs) with the Pose Graph Model (PGM) for tasks related to HPE. These approaches utilize a self-independent Pose Graph Model (PGM) applied to the score maps produced by DCNNs [13,14,15]. The PGM parameters are learned during the end-to-end training process. Subsequently, these parameters are fixed and uniformly applied to all input data during the prediction.

Attention mechanisms have significantly advanced deep learning, particularly in enhancing focus on crucial data segments. Visual attention, a key application of this concept, has shown remarkable efficacy across various domains, including image and text processing. In this study, we present the Spatial Deep Feature Pose Graph Network (SDFPoseGraphNet), building upon our previous work [16]. This network extends our previous approach by integrating the self attention (SA) module with the VGG-19 model in the backbone. The SA module enables the network to dynamically prioritize important spatial regions in the input image that are crucial for accurate pose estimation. This integration aims to enhance the model’s feature representation capacity, leading to improved performance in HPE tasks. To address the challenge mentioned above for accurate HPE, we incorporate a PGM into the SDFPoseGraphNet. The PGM utilizes Second Inference Module (SIM) parameters that are adaptively learned from the deep feature maps extracted by VGG-19. This adaptivity allows the parameters to be tailored to the characteristics of each individual input image, leading to improved performance in pose estimation. In addition, our model utilizes the First Inference Module (FIM) potentials, which are score maps indicating the location of each hand joint, obtained from another module of the SDFPoseGraphNet. These FIM potentials, combined with the SIM parameters, contribute to the final pose estimation by the PGM. The inference process involves techniques like message passing, which enable the model to refine and enhance the accuracy of the joint predictions. A significant advantage of the SDFPoseGraphNet is its end-to-end trainable nature, where all components, including VGG-19 and the PGM, can be jointly optimized during the training process. This general optimization ensures that the deep feature maps extracted by VGG-19 via spatial attention (SA) [17] and the adaptively learned parameters of the PGM work well to achieve precise and reliable HPE.

Our contribution is three-fold and is stated below as follows:We introduce SDFPoseGraphNet, a novel framework that enhances the capabilities of VGG-19 by incorporating SA mechanisms; our model effectively captures spatial information from hand images, allowing VGG-19 to extract deep feature maps that capture intricate relationships among various hand joints.To address the challenge of accurate pose estimation, we incorporate a PGM into SDFPoseGraphNet. This model utilizes adaptively learned SIM parameters, derived from the deep feature maps extracted by VGG-19, to model the geometric constraints among hand joints. The adaptivity of the parameters enables personalized pose estimation, tailoring the model to the unique characteristics of each individual input image.The model combines FIM potentials and SIM parameters, which play a crucial role in the final pose estimation performed by the PGM. The inference process incorporates techniques like message passing, refining, and enhancing the accuracy of the joint predictions.

This article follows a structured approach with several sections. Section 2 presents an overview of prior research conducted in the same field. Section 3 elaborates on the comprehensive methodology of our proposed model. Section 4 covers pertinent information regarding the experimental setup and implementation details. The results and analysis are presented in Section 5, while in Section 6, we conclude our work.

## 2. Related Works

HPE is a complex task that faces several challenges, including variations in hand poses, limited depth information, and issues related to appearance and occlusion. Researchers have been actively exploring different approaches and techniques to tackle these challenges. One effective approach is the use of multi-view RGB models [4,18,19,20], which address the problem of self/object occlusion and achieve notable accuracy in HPE. However, the practical implementation of multi-view RGB models [21,22] is constrained by the requirement for specific camera setups, limiting their performance in real-world scenarios. Depth-based models [23,24,25] offer advantages in accurately locating and identifying hands based on depth values, resulting in faster processing. However, these models can be sensitive to lighting conditions and noise, making them more suitable for controlled environments where such factors can be regulated. In recent years, RGB cameras have been widely adopted in HPE tasks due to their anti-interference capabilities, affordability, and portability. The research community has been actively developing methods to directly estimate 3D hand poses from RGB images. Some approaches involve fitting 3D models using estimated 2D joint locations. It is important to note that the accuracy of 2D HPE greatly influences the overall performance of 3D HPE techniques [9,26].

CNNs are crucial for HPE [2,27], leveraging their exceptional feature extraction capabilities. The automatic learning of feature representations in modern deep learning eliminates the need for manual feature engineering. However, the quality of learned features relies on the network architecture used. Thus, exploring network design methods becomes imperative for extracting tailored features for accurate hand pose estimation. These methods aim to identify optimal configurations that extract highly discriminative and informative features, enhancing system accuracy and robustness. Some research work focuses on directly mapping the input image to the coordinates of keypoints in a 2D or 3D space, known as holistic regression [28,29]. This approach eliminates the need for intermediate representations, such as pixel-wise classification, while capturing global constraints and correlations between keypoints. However, concerns have been raised regarding the generalization capability of holistic regression and its sensitivity to translational variance, which can lead to diminished prediction accuracy [30]. CPM, which enforces CNN to generate heatmaps, indicates the location of keypoints. Heatmap-based methods have resulted in a significant increase in performance; therefore, we followed the heatmap-based approach for precise prediction of the keypoints pertaining to each joint of the hand.

## 3. SDFPoseGraphNet

The CPM, although capable of learning strong feature maps, often encounters difficulties in capturing the geometric relationships between joints. As a result, the final predictions of joint positions can exhibit inconsistencies, which pose a significant challenge in tasks involving human pose estimation. This challenge becomes even more prominent in 2D HPE due to the higher level of articulation and the presence of self-occlusion, which further exacerbates the problem. To solve these limitations, we propose SDFPoseGraphNet, a novel framework that enhances the capabilities of VGG-19 with SA, visually depicted in Figure 1.

The proposed model incorporates two modules called the First Inference Module (FIM) and the Second Inference Module (SIM). A final Graphical Inference Module is employed to connect the FIM and SIM sequentially. The FIM produces a provisional feature score for the keypoints *K* of the hand during the preliminary stage. The VGG-19 block’s revert feature score can readily integrate with the proposed SDFPoseGraphNet. The final module utilizes the parameters generated by the SIM, which depict spatial constraints among the critical points of the hand. The suggested framework for HPE differentiates itself from previous frameworks by leveraging the SIM, which enables the association of the final graphical module with DCNN [13,14]. In our work, the parameters of the PGM are not considered as independent parameters. Rather, they are tightly linked to the input image through VGG-19, making them adaptable and responsive to different variations in the input images. This integration allows the model to capture and utilize relevant information from the input image effectively, resulting in improved performance and adaptability in various circumstances, as mentioned earlier.

The process of predicting hand poses can be formally described through a graph represented by G, which consists of vertices V and edges E. In other words, it can be denoted as G=(V,E). Here, the vertices V are directly associated with the salient keypoints of the hand, denoted as *K*, and can be expressed as $V=\left\{v_{1},v_{2},\ldots ,v_{K}\right\}$. Each vertex $v_{i}$ corresponds to a specific two-dimensional keypoint, represented as $x_{i}\in R^{2}$, which provides the position of that keypoint relative to $v_{i}$. The subsequent equation, labeled as (Equation 1), expresses the joint probability of the hand poses.
(1)$$p\left(X\setminus I,\Theta \right)=\;\frac{1}{Z}\prod_{i=1}^{\left|V\right|}\phi_{i}\left(x_{i}\setminus I;\Theta_{f}\right)\prod_{(i,j)\in E}\varphi_{i,j}\left(x_{i},x_{j}\setminus I;\Theta_{s}\right)$$
In this equation, $X=\left\{x_{1},x_{2},\ldots ,x_{K}\right\}$ represents the set of hand keypoints, and *i* and *j* denote their positions. The term |V| signifies the cardinality, or the number of elements, in the set V. Z and I correspond to the input image. The parameter Θ encompasses the combination of FIM and SIM; $\Theta =\phi_{i}\left(x_{i}\setminus I;\Theta_{f}\right);\varphi i,j\left(x_{i},x_{j}\setminus I;\Theta_{s}\right)$.

In this context, the equation models the joint probability of hand poses by considering the interrelationships between keypoints and their positions within a graphical structure. It provides a formal representation of the predictive process used to estimate hand poses, where various components, such as the hand keypoints and the input image, are considered in the calculation of this probability. The parameter Θ encapsulates the combination of specific modules that contribute to the overall predictive model.

Further information and comprehensive explanations regarding each component of the proposed model SDFPoseGraphNet are introduced in the following subsections.

### 3.1. Optimized VGG-19 Backbone with SA Module Integration for Improved 2D HPE Feature Extraction

In a neural network, attention systems enhance focus on crucial portions of the input data while diminishing the importance of less relevant components. This significant approach, known as visual attention, has made substantial strides in the realm of DL research. These attention mechanisms have shown efficacy in dealing with both text and image data. To augment the existing efficiency of convolutions, numerous methods incorporating visual attention have been developed. This study proposes an innovative approach for HPE tasks by integrating the SA module with the VGG-19 model. This fusion leverages the known effectiveness of the VGG-19 architecture in feature extraction and the ability of attention mechanisms to zero in on salient spatial regions, thus creating a potent combination [31]. These SA modules allow each feature map to implement a distinct SA mechanism. Attention maps created this way are assembled along the channel dimension and are then passed through a convolutional layer with a kernel size *k* to yield the final attention map. To ensure that the values lie between 0 and 1, this final attention map is passed through a sigmoid activation function, which normalizes the range. These normalized values indicate the significance of each spatial location in the model.

VGG-19 produces five distinct feature maps $\left(f_{1},\ldots ,\;f_{4},f_{5}\right)$, as shown in Figure 2. SA $\left(s_{1},\ldots ,\;s_{4}\right)$ and 1×1 were implemented on the first four feature maps, along with 1×1 convolution for channel reduction, to obtain the attention maps as formulated in Equations (Equation 1) and (Equation 2).
(2)$$F_{i}=s_{i}\left(f_{i}\right)$$

Here, $F_{i}$ is the attention map after applying the spatial attention module $s_{i}$ to feature map $f_{i}$ from VGG-19.
(3)$${\hat{F}}_{i}=C\left(f_{i}\right)$$

${\hat{F}}_{i}$ is the feature map after channel reduction, and *C* denotes a convolution operation. This series of convolutional layers reduces the channel dimension of the feature maps to a value of 128.
(4)$$F_{i}={\hat{F}}_{i}⊗F_{i},$$
where ⊗ indicates element-wise multiplication, and $F_{i}$ represents the attention feature map. To align the spatial dimensions, the interpolation process is executed by utilizing bilinear interpolation (F.interpolate) from torch.nn.functional, as shown in Equation (Equation 5).
(5)$${\hat{S}}_{i}=F.interpolate\left(F_{i},\;t\right)$$

${\hat{S}}_{i}$ represents attention maps after interpolation, and *t* is the target size $f_{5}$, which is the last feature map without spatial attention. After interpolation, we fused the attention maps with the last feature map from the backbone by element-wise addition, as shown in Equation (Equation 6):(6)$$S={\hat{S}}_{1}+\ldots +{\hat{S}}_{4}+f_{5},$$
where S is the fused attention map.

### 3.2. Architectural and Operational Insights into the FIM

In the initial module, the VGG-19 architecture was used up to Conv 3×3 as the fundamental feature extraction network followed by three convolutional layers that facilitated the generation of the initial heatmap. The VGG-19 architecture was enhanced by incorporating an SA module, generating 128 feature maps. The produced feature maps undergo information processing through a module that consists of six stages. These stages involve a series of continuous convolution layers with a specific kernel size 3×3, each incorporating a heatmap label as a supervisory mechanism. The generation of these labels was achieved by applying a Gaussian function to the corresponding ground truth, which is shown below.
(7)$$Heatmap=exp(\frac{-[{\left(x-x_{k}\right)}^{2}+{\left(y-y_{k}\right)}^{2}]}{2\delta^{2}})$$

The symbol δ denotes the extent of the heatmap, while $x_{k}$ and $y_{k}$ represent the underlying coordinates on the ground. The final stage produces 21 unique feature maps that serve as a representation of each keypoint. These feature maps are utilized as static weights during the training of the graphical module. The output of the initial module is denoted as $F\;\left(I;\Theta_{f}\right)\in R^{\left|V\right|\times h_{F-heatmap}\times w_{F-heatmap}}$. It is important to note that the dimensions of the output heatmaps are determined by the corresponding values of the height $h_{F-heatmap}$ and width $w_{F-heatmap}$. Figure 3 shows the sequential procedure for convolutional detailing of the FIM.

### 3.3. Architectural and Operational Insights into the SIM

The SIM follows a similar methodology as the FIM. In particular, it upholds the most recent framework for 128 feature maps, but it generates 40 feature maps instead of 21 feature maps. The 40 channels or feature maps generated by the SIM represent information about the relationships between hand keypoints. These feature maps capture and encode information about the relative positions, distances, and interactions between different pairs of keypoints on the hand. The result produced by SIM is represented as $S\left(I;\Theta_{S}\right)\in R^{\left|E\right|\times h_{s}\times w_{s}}$, which is indicative of the SIM channel kernels. The purpose of the SIM is to learn the relative positions between hand keypoints.

When training the SIM, we maintain the weights of the FIM in a fixed state, effectively ’freezing’ them. As depicted by the gray arrows in Figure 1, there is a directional flow of information originating from the FIM that moves towards the SIM at the end of each stage. During this information flow process, the feature sets generated at each FIM stage are merged with those from the corresponding SIM stages. For instance, the features from the first stage of the FIM are combined with those from the first stage of the SIM, and this composite feature set is then fed into the second stage of the SIM. This method of consistent information exchange is applied throughout all the stages of the SIM’s training phase. The detailed structure of this architecture is illustrated in Figure 3.

### 3.4. Final Graph Inference Module

The message-passing algorithm is widely utilized in graphical model inference. It typically facilitates the effective calculation of marginal probabilities by using the sum-product operation within a graphical module. The equation for the marginal probability can be expressed in Equation (Equation 8).
(8)$$p_{i}\left(x_{i}\setminus I;\Theta \right)=\sum_{V\setminus x_{i}}p(X\setminus I;\Theta )$$

In this scenario, the function of argmax probability is employed to optimize the marginal probability for predicting the location of the hand keypoint labeled as *i*:(9)$$x_{i}=argmax\;p_{i}\left(x_{i}\setminus I;\Theta \right),$$
where $\Theta =\{\Theta_{f},\Theta_{s}\}$ is the collection of all parameters, which amalgamate the parameters of the initial two modules. In the graphical model, each vertex V has the ability to both send and receive messages M to and from its corresponding neighboring nodes Nbn. The sum-product algorithm is responsible for updating the messages sent from hand keypoints from *i* to *j*. The complete message exchange is denoted by $M_{ij}$; here, $M_{ij}\in R^{h_{w}\times w_{u}}$ depicts the entire formulation of the message passing.
(10)$$M_{ij}\left(x_{j}\right)=\sum_{x_{i}}\varphi_{i,j}(x_{i}\;,x_{j})\phi_{i}\left(x_{i}\right)\prod_{k\in Nbn(i)j}m_{ki}\left(x_{i}\right)$$

Given that $x_{j}$ assumes values from a grid point set with dimensions of $h_{w}\times w_{u}$, we approximate the marginal probabilities as follows, after multiple iterations and upon convergence:(11)$$p\left(x_{j}\right)\approx \;\frac{1}{Z}\phi_{i}\left(x_{i}\right)\prod_{k\in Nbn(i)}m_{ki}\left(x_{i}\right),$$
where $m_{ki}\left(x_{i}\right)$ represents a message from node *k* to node *i*, and Z is the normalization.

This research work adopted a tree-structured graphical model. One significant advantage of this model is the ability to accurately derive the marginal probability outlined in the earlier equation using belief propagation. Figure 4 illustrates the hand model arranged in a tree-like structure. Precise marginals can be determined by transmitting messages from the bottom-most nodes to the topmost node, and then back down to the lowest nodes. Numerical values are presented next to each arrow in Figure 4 with the schedule of message updates denoted by the number 3. In total, 40 message transmissions are adequate for obtaining accurate marginals.

## 4. Experimental Setup

### 4.1. Dataset

The Carnegie Mellon University (CMU) Panoptic Hand Dataset was utilized during our study to assess the proposed model. The dataset consists of a total of 14,817 annotations that correspond to the right hand of individuals captured in images from the Panoptic Studio. The current research examines the process of HPE, as opposed to hand detection. To achieve this objective, annotated hand image patches were extracted from the initial images using a square bounding box with dimensions 2.2 times larger than the hand size. The dataset was partitioned into three subgroups using a random sampling technique as shown in Table 1. Specifically, these subgroups were designated as: the training set, comprising 80% of the data; the validation set, comprising 10% of the data; and the test set, comprising 10%.

### 4.2. Implementation Details

The proposed model is implemented using the PyTorch framework version 1.11.0+cu102. The present model underwent a tripartite training process, where each stage was trained with a consistent learning rate of $1\times 10^{-4}$, a batch size of 32, and four workers. The first two stages of the model were trained for 100 epochs, and an early stop technique was implemented to mitigate overfitting. In contrast, the last stage was trained for a notably shorter duration of 10 epochs with a weight decay of 0.01.

### 4.3. Loss Function

The mean squared error (MSE) is utilized as the loss function in the model. To prevent the diminution of the loss from reaching nominal values, the loss function is scaled by a coefficient of 35.

The formulation of the loss calculation for a model involves a weighted sum of the loss function of each inference.
(12)$$L=\alpha_{1}L^{First}+\alpha_{2}L^{Second}+\alpha_{3}L^{Final}$$

$L^{First}$ is the MSE loss of FIM, $L^{Second}$ represents the MSE loss of SIM, and $L^{Final}$ denotes the PGM MSE loss. These loss terms collectively drive the training process for enhanced model performance. While $\alpha_{1}$, $\alpha_{2}$, and $\alpha_{3}$ are the coefficients for fine-tuning the model, the values are set to 1, 0.1, and 0.1, respectively.

### 4.4. Model Optimization

An optimizer aims to decrease the loss function and steer the network toward improved performance by identifying optimal parameter values. Utilizing a newly derived variation of the Adam optimizer called AdamW can bolster the refinement of model optimization techniques. In contrast to its predecessor, the Adam optimizer, the AdamW algorithm effectively disentangles the weight decay component from the learning rate, allowing for individualized optimization of each component. This feature effectively addresses the issue of excessive overfitting. The outcomes reveal that the models optimized through AdamW exhibit superior generalization performance compared to those trained using other optimizers, particularly Adam. The AdamW optimizer was employed in the training of our final graphical module.

### 4.5. Activation Functions

Several activation functions, namely ReLU, SoftMax, and Mish, introduce nonlinear components to the neural network, allowing it to comprehend complex patterns and correlations in the data. The Mish activation function has demonstrated superior performance to alternative activation functions, primarily due to its nonlinear nature [32]. The definition of the term can be expressed using the following formula: (13)$$f\left(x\right)=xtanh(ln\left(1+e^{x}\right)).$$

The experimental findings demonstrate that the efficacy of Mish surpasses that of widely utilized activation functions, including ReLU and SoftMax, among others, in diverse deep network architectures operating on complex datasets.

### 4.6. Evaluation Metric

We normalized the Percentage of Correct Keypoints (PCK) [4] for our study. The PCK metric [4] is a commonly employed evaluation measure for HPE. Specifically, it quantifies the likelihood that a predicted keypoint is located within a designated distance threshold, denoted as σ, from its corresponding ground truth coordinate. The application of σ, restricted to the scale of the hand bounding box, is utilized within this study. The threshold was uniformly distributed within the range of 0 to 0.16, and the PCK formula is
(14)$$PCK_{\sigma }^{k}=\frac{1}{\left||D|\right|}=\sum_{D}1(\frac{\left|\right|p_{k}^{pt}-p_{k}^{gd}{\left|\right|}_{2}}{max(w,h)}\leq \delta ),$$
where $p_{k}^{gd}$ is the ground truth of the keypoint, 1 is the indicator function, and $p_{k}^{pt}$ is the predicted keypoint. *k* represents the number of keypoints, *D* represents the number of test or validation samples, and *h* and *w* represent the height and width of the sample images, respectively.

## 5. Results and Analysis

The subsequent section of this paper illustrates an exhaustive performance analysis of the proposed SDFPoseGraphNet. By undertaking a series of ablation studies, we thoroughly scrutinize the functioning of each module. Further, a comparative study is conducted between our proposed network and traditional networks employed for HPE. Finally, the predicted outcomes are made visually comprehensible to underline our results.

### 5.1. Quantitative Results

Table 2 numerically illustrates the PCK performance of our proposed model on the CMU Panoptic Hand Dataset. Upon examination, it is evident that our SDFPoseGraphNet surpasses other contemporary state-of-the-art models. The empirical results indicate that the SDFPoseGraphNet, on average, enhances precision by 7.49% when compared against the CPM [12] results. Furthermore, it improves accuracy by nearly 3.84% in comparison with AGMN [33], and 1.71% in relation to CDGCN [16]. We provide a comprehensive breakdown of the numerical results for each module, i.e., the FIM, SIM, and our proposed SDFPoseGraphNet, in Table 3. Additionally, Figure 5a,b presents the likelihood of correct keypoints of our proposed model in relation to other validation and test data extracted from the CMU Panoptic Hand Dataset.

### 5.2. Qualitative Results

To demonstrate qualitative results, we chose a diverse selection of images featuring different angles, complicated situations, instances of self/object occlusion, and complex backgrounds. Figure 6 displays the high resilience and consistency of SDFPoseGraphNet, as reflected in its reliable performance across myriad test scenarios and conditions. Its anti-interference capability managed to perform efficiently even amidst complex backgrounds. In situations where the image clarity was compromised, obtaining a high-resolution version or a more detailed depiction was deemed necessary for improved interpretation and analysis. This context underscores the relevance of our proposed model. Figure 7 visually represents our model’s performance on a random selection of images. The figure includes: (a) ground truth, (b) our proposed SDFPoseGraphNet model, (c) CDGCNN, and (d) AGMN.

### 5.3. Ablation Study

An ablation study was carried out using the Panoptic dataset to substantiate the effectiveness of our optimization strategy. To evaluate the impact of the SA module, it was integrated into the FIM, while all other aspects were kept unchanged. As per the experimental results demonstrated in Table 4, an average performance enhancement of 3.52% was observed. Figure 8a illustrates the notable improvement in the network’s output by integrating VGG-19 with SA. The SIM generates 40 feature maps as explained above. However, SIM consistently predicts the same 2D coordinates for pairs of neighboring hand keypoints that share a common edge in the tree structure. This consistency arises because the relative positions between these keypoints are fixed and learned during training. Consequently, during testing, the SIM consistently predicts the same relative positions, resulting in consistent 2D coordinate predictions for these 21 keypoints.

We also utilized a VGG-19 backbone model with a few extra layers and batch normalization for feature enhancement. Our analysis of each module in Table 5 shows an average improvement of 2.48% and 0.39% over AGMN [33] and CDGCN [16], respectively. Performance in terms of accuracy increases, while computational speed decreases due to the added layers.

The presence of noise within a dataset can significantly reduce the model’s performance [34]. Considering this statement, we preprocessed the CMU Panoptic dataset using a median filter. Later, the model was trained using the process, and the outcomes are presented herein. It has been observed that the implementation of denoising filters on the dataset results in a level of smoothness that can significantly diminish the clarity and discernibility of edges, thereby presenting a suboptimal approach to hand pose estimation. As illustrated in Figure 9, it is evident that the preprocessed image exhibits a considerable degree of smoothness relative to the original image, causing information loss. However, at some locations where the noise exists, the model tends to perform better after the removal of noise. On the other hand, some points in the images may not contain noise, affecting the model’s performance.

Table 6 and Table 7 show the numerical results of each module with preprocessed data. The quantitative results show that our proposed model performs better than the existing methods presented in [16,33] with processed and unprocessed data, even though we noticed that the median filter applied on the dataset causes an information loss. Figure 8b illustrates the PCK with original and preprocessed data in the test dataset.

## 6. Conclusions

This research proposed the SDFPoseGraphNet, a novel framework that combines the power of VGG-19 and spatial attention to improve hand pose estimation. By leveraging the deep feature maps extracted by VGG-19, the model captures spatial information and learns the relationships among hand joints. The incorporation of a Pose Graph Model with dynamically learned SIM parameters further enhances accuracy. FIM potentials obtained from another module contribute to pose estimation, and message-passing techniques refine joint predictions. The SDFPoseGraphNet is end-to-end trainable, allowing for joint optimization of all components, resulting in precise and reliable hand pose estimation. The present model possesses the attribute of generality, rendering its potential applicability useful in various forthcoming computer vision undertakings, such as the estimation of 3D hand poses and human poses, among others.

## Figures and Tables

**Figure 1 sensors-23-09088-f001:**
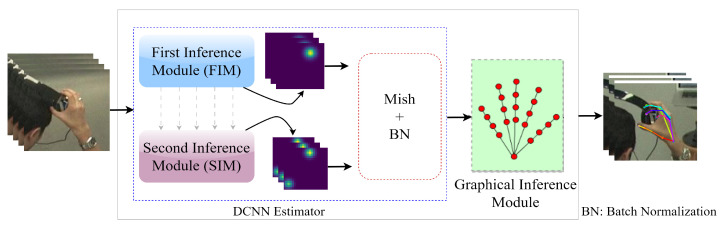
Illustration of the SDFPoseGraphNet architectural design.

**Figure 2 sensors-23-09088-f002:**
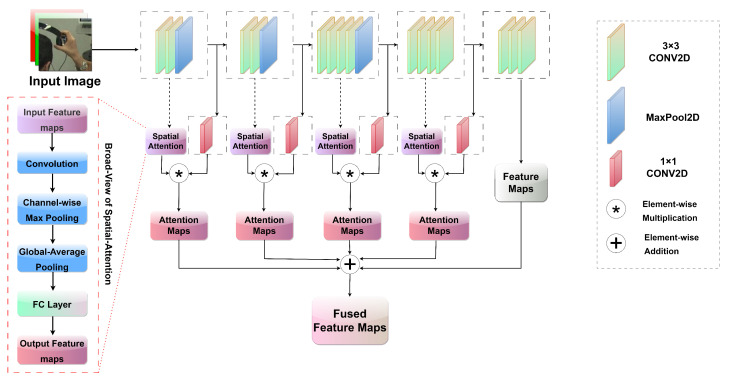
Architecture of VGG-19 with SA module for enhanced 2D HPE.

**Figure 3 sensors-23-09088-f003:**
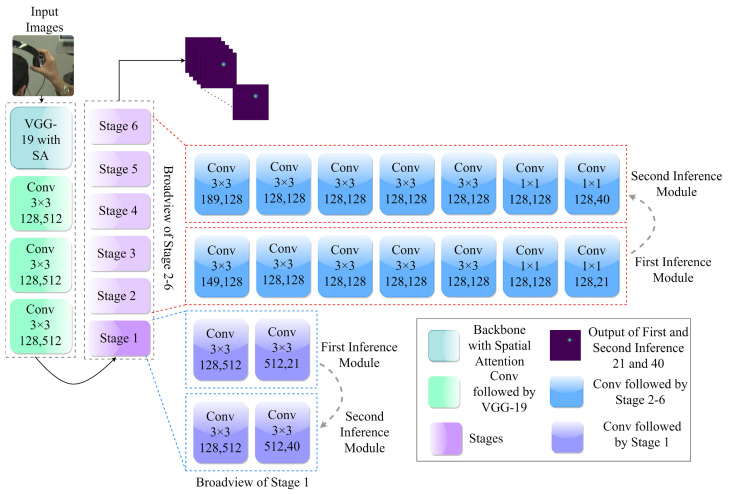
Comprehensive overview of First and Second Inference Modules (FIM and SIM).

**Figure 4 sensors-23-09088-f004:**
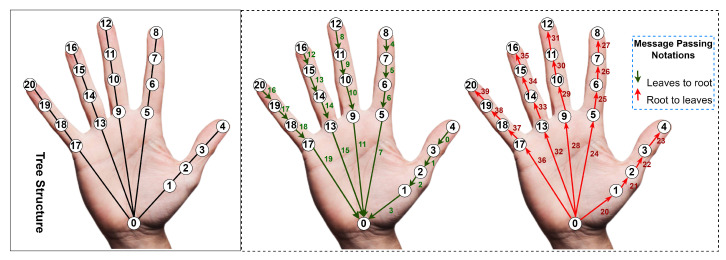
Illustrative representation of message passing within a hand tree structure.

**Figure 5 sensors-23-09088-f005:**
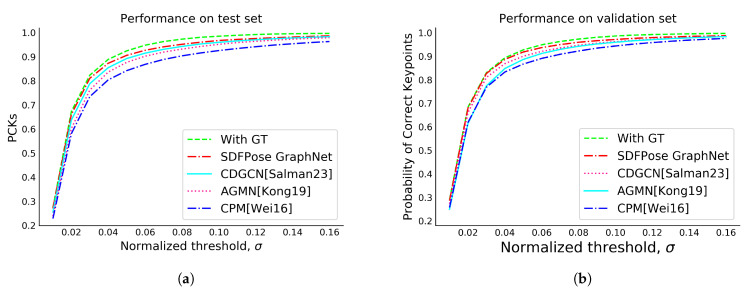
PCK evaluation for performance comparison: proposed model against existing models [12,16,33]. (**a**) Test; (**b**) Validation.

**Figure 6 sensors-23-09088-f006:**
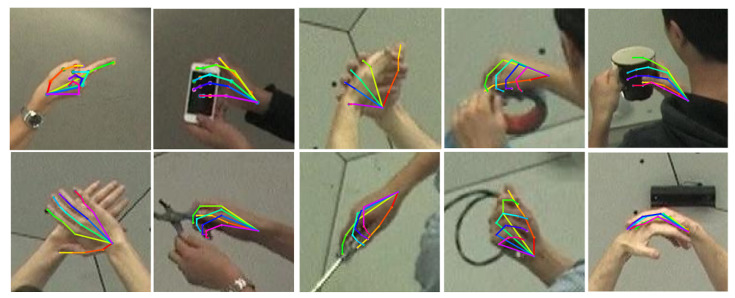
Visualizing the performance of SDFPoseGraphNet: random image analysis.

**Figure 7 sensors-23-09088-f007:**
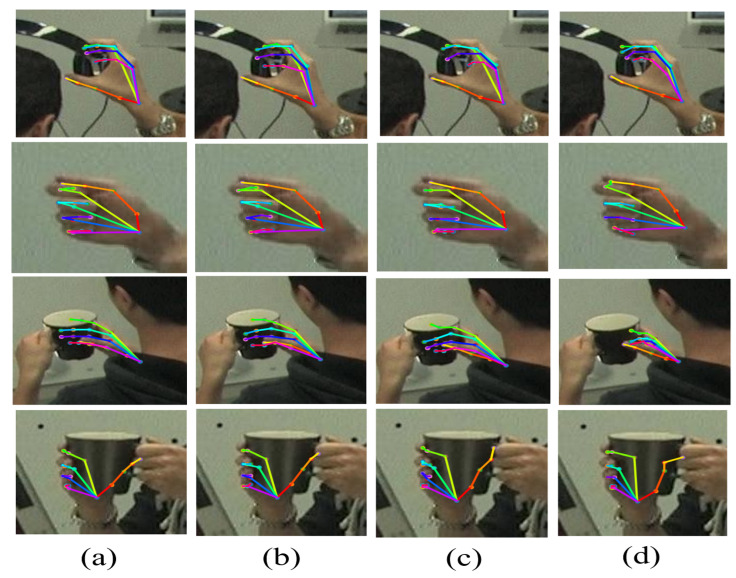
Illustrative comparison of 2D HPE: (**a**) Ground truth; (**b**) Ours; (**c**) CDGCN [16]; and (**d**) AGMN [33].

**Figure 8 sensors-23-09088-f008:**
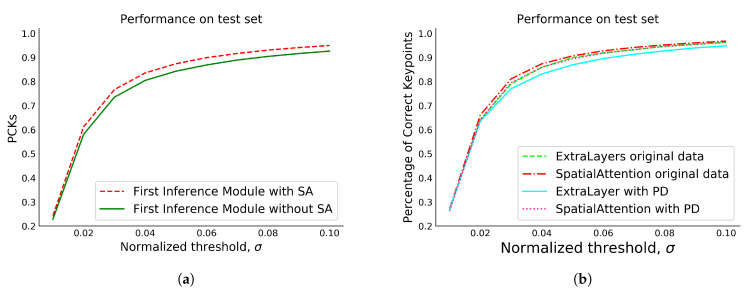
(**a**) PCK comparison of the First Inference Module (FIM) with and without the integration of the SA module; (**b**) PCK comparison of FIM with preprocessed data (PD) and original data.

**Figure 9 sensors-23-09088-f009:**
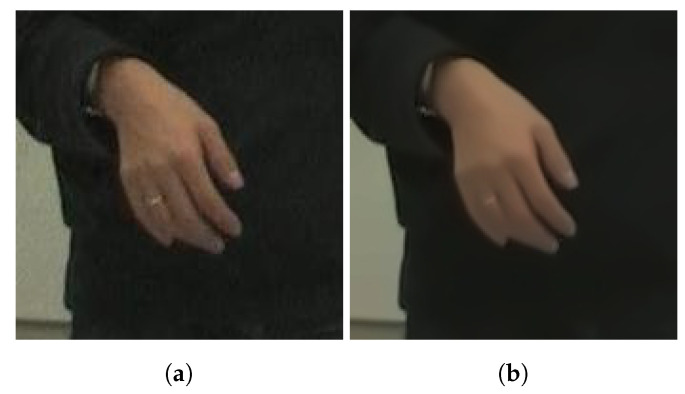
(**a**) Original image (**b**) Preprocessed image.

**Table 1 sensors-23-09088-t001:** Distribution of data.

Dataset	Training	Validation	Testing
CMU Panoptic	11,853	1482	1482

**Table 2 sensors-23-09088-t002:** SDFPoseGraphNet performance in comparison with previous state-of-the-art models.

Threshold σ	0.01	0.02	0.03	0.04	0.05	0.06	0.07	0.08	0.09	0.10	Average
CPM [12]	22.88	58.10	73.48	80.45	84.27	86.88	88.91	90.42	91.61	92.61	76.96
AGMN [33]	23.90	60.26	76.21	83.70	87.70	90.27	91.97	93.23	94.30	95.20	79.67
CDGCN [16]	25.60	63.77	78.90	85.52	89.30	91.53	93.12	94.33	95.33	96.02	81.34
SDFPose GraphNet	**27.11**	**66.12**	**81.10**	**87.34**	**90.62**	**92.73**	**94.19**	**95.21**	**96.07**	**96.79**	**82.73**

**Table 3 sensors-23-09088-t003:** Comprehensive breakdown of numerical results: module-wise performance analysis.

Threshold σ	0.01	0.02	0.03	0.04	0.05	0.06	0.07	0.08	0.09	0.10
FIM	24.28	61.21	76.63	83.55	87.36	89.90	91.64	93.03	94.08	94.97
SIM	24.85	62.25	78.18	85.47	89.33	91.60	93.17	94.41	95.50	96.26
SDFPose GraphNet	**27.11**	**66.12**	**81.10**	**87.34**	**90.62**	**92.73**	**94.19**	**95.21**	**96.07**	**96.79**

**Table 4 sensors-23-09088-t004:** Comparative performance evaluation of FIM with and without SA integration.

Threshold σ	0.01	0.02	0.03	0.04	0.05	0.06	0.07	0.08	0.09	0.10
FIM	22.88	58.10	73.48	80.45	84.27	86.88	88.91	90.42	91.61	92.61
FIM with SA	24.28	61.21	76.63	83.55	87.36	89.90	91.64	93.03	94.08	94.97

**Table 5 sensors-23-09088-t005:** Module performance comparison with integrated extra feature extraction layers.

Threshold σ	0.01	0.02	0.03	0.04	0.05	0.06	0.07	0.08	0.09	0.10
FIM	24.53	60.82	75.84	82.72	86.47	89.07	90.98	92.42	93.52	94.44
SIM	23.85	60.11	76.21	83.68	87.87	90.52	92.44	93.84	94.85	95.63
SDFPose GraphNet	26.25	64.22	79.44	85.93	89.41	91.74	93.30	94.51	95.42	96.22

**Table 6 sensors-23-09088-t006:** Comparative performance metrics of the enhanced model with preprocessed data and additional feature layers.

Threshold σ	0.01	0.02	0.03	0.04	0.05	0.06	0.07	0.08	0.09	0.10
FIM	25.43	61.30	75.30	81.48	85.02	87.56	89.45	90.80	91.96	92.97
SIM	23.97	60.26	76.03	83.29	87.21	89.86	91.84	93.14	94.21	94.99
SDFPose GraphNet	26.75	63.57	77.01	83.13	86.91	89.52	91.28	92.72	93.96	94.79

**Table 7 sensors-23-09088-t007:** Comparative analysis of model performance with preprocessed data and SA integration.

Threshold σ	0.01	0.02	0.03	0.04	0.05	0.06	0.07	0.08	0.09	0.10
FIM	25.90	62.87	76.64	82.77	86.33	88.54	90.37	91.63	92.79	93.74
SIM	24.38	61.67	77.71	84.69	88.59	90.98	92.69	93.92	94.87	95.71
SDFPose GraphNet	26.25	64.12	79.01	85.89	89.89	91.88	93.14	94.67	95.44	96.33

## Data Availability

Data are contained within the article.

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
