# Peer review of "SDFPoseGraphNet: Spatial Deep Feature Pose Graph Network for 2D Hand Pose Estimation"

_sensors, 2023, doi:10.3390/s23229088_

Round 1

Reviewer 1 Report

Comments and Suggestions for Authors

The manuscript presents a novel architecture for hand pose estimation. The paper is well written and the results look good. 

Given that the VGG architecture is slower compared to more recent architectures, I would like to see a comparison in terms of inference time compared to state of the art models.

Author Response

I have solved the mentioned comments according to my best knowledge and understanding. Please find the attachment below. 

Reviewer 2 Report

Comments and Suggestions for Authors

The manuscript is generally well composed. Just a couple of comments and suggestions: 1. There are unexplained abbreviations in Lines 13, 16-17, and 131. Although some have been explained in the Abstract, please explain them again when they appear for the first time in the main text body. 2. Equation (1) needs a better explanation, especially the meanings of the geometric variables contained in the combined parameter. Please use a diagram with those variables labeled clearly.

Comments on the Quality of English Language

Please use the past tense for the activities performed in the past. 

Author Response

(The authors gave the same response as above.)

Reviewer 3 Report

Comments and Suggestions for Authors

Summary:

In this manuscript, the authors propose a method (Spatial Deep Feature Pose Graph Network) for hand pose estimation on 2D images on CMU Panoptic hand dataset. The results show some improvements compared to previous methods.

Comment:

- The authors describe several challenges in hand pose estimation, but they are not consistent throughout the text, please describe the challenges the manuscript addresses and make it consistent in text.

- Most of the reference in the related work are not recent, authors need to add more reviews on more recent work.

- The description of method is ambiguous and unclear. Start with the description of the overall figure, then describe each component/module in order, not the order is difficult for the readers to understand.

- This work is based on author's previous work, the authors should explicitly mention what has been further developed based on previous work.

Detailed comment:

- Line 175, F_hat_ii should be F_hat_i.

- Section 3.3, what exactly is the first stage and second stage? What are the six stages in the second stage?

- Why SIM outputs 40 feature maps? Are they also related to keypoints? How is the information from FIM flows to SIM? I don't see gray arrows in fig 1. Pleas elaborate.

- Conceptually, what is the purpose of SIM?

- What are the first, second and final loss term in loss function L? Please explain.

- In ablation study, please specify how the authors ablate SIM. The feature maps of SIM output are not correspondent to keypoint.

Author Response

(The authors gave the same response as above.)

Round 2

Reviewer 1 Report

Comments and Suggestions for Authors

Please compare the inference times using the times reported in their respective papers.

Author Response

Hello Sir/Mam,

Hope you are doing well, we tried to respond to each comment that we got to our best knowledge and understanding. Please find the attachment below

Thank you.

Reviewer 3 Report

Comments and Suggestions for Authors

After reading the authors' responses, I can see some of my previous comments are addressed, but some comments remain unaddressed.

Comments:

- First of all, it is difficult to read the revised manuscript without highlighting what has been changed. I suggest the authors highlight the changes.

- Point 4 is still not addressed, what I meant by 'explicitly mention what has been further developed based on previous work' was that, what was the method in previous work, what has been newly developed in this work and what motivates the changes, etc. These should be explicitly stated.

- Point 7, the gray arrows in Figure 1 do not reveal how exactly the information flows. I'd also add the details of the information flow (are the flows one-to-one between conv block or else?) in Figure 3 and describe how those features are processed from FIM to SIM.

- Point 9, the loss function is still unclear, I need to see 1) Are L_first, L_second and L_final all MSE? 2) What are the predictions and labels in each loss term? 3) The weights of each loss term (alpha_1, alpha_2, alpha_3), are they all 35 or else.

Author Response

(The authors gave the same response as above.)
